# The Role of Sirtuin 3 in Radiation-Induced Long-Term Persistent Liver Injury

**DOI:** 10.3390/antiox9050409

**Published:** 2020-05-11

**Authors:** Francesca V. LoBianco, Kimberly J. Krager, Gwendolyn S. Carter, Sinthia Alam, Youzhong Yuan, Elise G. Lavoie, Jonathan A. Dranoff, Nukhet Aykin-Burns

**Affiliations:** 1Division of Radiation Health, University of Arkansas for Medical Sciences, 4301 W. Markham St., Little Rock, AR 72205, USA; FVLobianco@uams.edu (F.V.L.); KJKrager@uams.edu (K.J.K.); GSCarter@uams.edu (G.S.C.); SAlam2@uams.edu (S.A.); 2Department of Pharmaceutical Sciences, University of Arkansas for Medical Sciences, 4301 W. Markham St., Little Rock, AR 72205, USA; 3Department of Pathology, University of Arkansas for Medical Sciences, 4301 W. Markham St., Little Rock, AR 72205, USA; YYuan@uams.edu; 4Department of Gastroenterology and Hepatology, University of Arkansas for Medical Sciences, 4301 W. Markham St., Little Rock, AR 72205, USA; EGLavoie@uams.edu (E.G.L.); JADranoff@uams.edu (J.A.D.)

**Keywords:** ionizing radiation, liver, sirtuin 3, hydroperoxide, inflammation

## Abstract

In patients with abdominal region cancers, ionizing radiation (IR)-induced long-term liver injury is a major limiting factor in the use of radiotherapy. Previously, the major mitochondrial deacetylase, sirtuin 3 (SIRT3), has been implicated to play an important role in the development of acute liver injury after total body irradiation but no studies to date have examined the role of SIRT3 in liver’s chronic response to radiation. In the current study, ten-month-old Sirt3^−/−^ and Sirt3^+/+^ male mice received 24 Gy radiation targeted to liver. Six months after exposure, irradiated Sirt3^−/−^ mice livers demonstrated histopathological elevations in inflammatory infiltration, the loss of mature bile ducts and higher DNA damage (TUNEL) as well as protein oxidation (3-nitrotyrosine). In addition, increased expression of inflammatory chemokines (IL-6, IL-1β, TGF-β) and fibrotic factors (Procollagen 1, α-SMA) were also measured in Sirt3^−/−^ mice following 24 Gy IR. The alterations measured in enzymatic activities of catalase, glutathione peroxidase, and glutathione reductase in the livers of irradiated Sirt3^−/−^ mice also implied that hydrogen peroxide and hydroperoxide sensitive signaling cascades in the absence of SIRT3 might contribute to the IR-induced long-term liver injury.

## 1. Introduction

The lifetime probability of males developing an invasive cancer is 42% and for females, 38%. Approximately 50% of all cancer cases benefit from radiation therapy [1]. More than 333,000 new cases of abdominal-region cancers are expected in the U.S. alone in 2020 [1], and a serious limitation to radiation therapy for these cancers is the possibility of irradiating the liver [2,3,4]. Ionizing radiation (IR)-induced liver toxicity remains a major challenge, especially for patients with underlying liver dysfunction, which is common with hepatic malignancies [3,4,5,6,7,8,9]. Clinical studies showed that IR-induced liver pathologies could be observed within 3 months of IR, even with IR doses as low as 30 Gy [10]. Previous clinical studies have shown patients can develop radiation damage within three months post-irradiation, which is caused by a veno-occlusive process. Fibrin accumulation of central veins and sinusoids followed by collagen deposition increases, resulting in increased portal pressure. These clinical manifestations have been termed “radiation-induced liver disease (RILD)”, which could progress to late-term radiation-induced fibrotic injury and eventual liver failure [4,11]. Received dose and volume of the liver that has been irradiated are two major parameters correlated with increased risk of developing RILD as well as its severity [12,13,14].

Stereotactic body radiation therapy (SBRT) could deliver high doses of IR to smaller volumes and provide survival benefits to those with liver and hepatobiliary cancers. However, conventional or SBRT IR doses delivered to the abdominal area, especially in patients with underlying liver dysfunction, are still constrained by concerns about RILD and hepatobiliary track toxicities [12,13,14,15,16]. The confounding factors that contribute to these pathologies are poorly understood, so determining the foremost molecular mechanisms is crucial for developing strategies to prevent and/or mitigate the side effects of radiotherapy [3,17,18].

Ionizing radiation can lead to oxidative events and biological activity changes in cells by directly interacting with target molecules, such as DNA, or through radiolysis of water. The biological activity alterations throughout the liver following radiation exposure are paramount because of the high mitochondrial content and oxidative metabolism [19,20,21]. Oxidative injury can arise from days to months after the initial exposure from the indirect generation of reactive oxygen species (ROS) and reactive nitrogen species (RNS), which are associated with chronic inflammatory responses [22,23,24]. Increased DNA damage, apoptosis and inflammation have been shown to play significant roles in RILD [4,13,25]. Several studies have also reported IR-induced mitochondrial dysfunction and oxidative stress in liver tissue that are thought to contribute to RILD [26,27,28]. Tao et al. (2010) and Coleman et al. (2014) demonstrated a causal link between ROS-mediated acute liver injury and decreased activity of manganese superoxide dismutase (MnSOD) in mice lacking sirtuin 3 (SIRT3) [29,30].

SIRT3 is the major mitochondrial nicotinamide adenine dinucleotide (NAD+)-dependent deacetylase. Mitochondrial deacetylation targets of SIRT3, such as peroxisome proliferator-activated receptor-gamma co-activator-1α, mitochondrial electron transport chain Complex I subunit NDUFA9, or Complex II subunit A, mitochondrial ribosomal protein L10 or TCA cycle enzyme isocitrate dehydrogenase 2, have been demonstrated to control a wide range of biological processes including but not limited to gene expression, metabolism, cancer and aging [30,31,32,33,34,35,36,37]. SIRT3 also plays a significant function as a stress response protein in regulation of ROS levels via posttranslational modification of antioxidant enzymes such as MnSOD and increasing their enzymatic activity. Therefore, it is logical to hypothesize that SIRT3, which is a pivotal mitochondrial stress-response protein, could also play a critical role in the long-term progression of liver toxicity seen in radiotherapy patients [38,39,40,41,42,43,44,45,46].

In an effort to best represent the age demographic most impacted in humans for developing liver and intrahepatic bile duct cancers, in the current study we used 10-month-old male mice for our liver only irradiation design. We followed the irradiated homozygous wild-type or Sirt3^−/−^ mice for an additional 6-month period for the development of long-term liver injury. Our results indicated significant increases in expression of inflammatory and profibrotic markers as well as decreased numbers of bile ducts in the irradiated areas of Sirt3^−/−^ livers compared to their sham treated controls. These mice also have more inflammatory cell infiltration in the portal and perivenular space. The DNA double strand breaks indicated by TUNEL staining and levels of protein oxidation in liver tissue were still elevated at the end of 6 months in Sirt3^−/−^ irradiated mice. Interestingly, there were no significant changes in MnSOD activity in sham versus irradiated groups in either Sirt3^−/−^ or Sirt3^+/+^ mice. However, activity peroxide-removing enzymes were significantly altered only in irradiated Sirt3^−/−^ livers, suggesting a shift in the types of reactive species contributing to the progression of long-term liver pathology following radiation exposure.

## 2. Materials and Methods

### 2.1. Image Guided Irradiation of Liver Tissue of Sirt3^+/+^ and Sirt3^−/−^ Male Mice

Littermates of Sirt3^+/+^ and Sirt3^−/−^ male mice on B6/Sv129 background were housed in a temperature and humidity controlled environment in filter top cages with *ad libitum* access to food and water at the University of Arkansas for Medical Sciences Animal Care facility until 10 months of age. Irradiation of the liver tissue was performed using the Small Animal Radiation Research Platform (SARRP, Xstrahl Inc., Suwanee, GA, USA) (Figure 1A). The mice were anaesthetized with 1% isoflurane inhalation for the duration of the radiation treatment. Each mouse was place supine on the horizontal mouse bed in the SARRP. A cone beam computed tomography (CBCT) image of each mouse was obtained to provide image guided radiation targeted to the liver at 60 kVp and 0.8 mA and reconstruction using 720 projections. From the image, precision targeting of the upper right lobe was determined. The liver was irradiated with 2 fractions of 12 Gy from a 90° and 0° gantry angle with a 7 mm and 5 mm tissue depth respectively. The liver treatments were performed utilizing a 0.5 mm copper filter with a 5 × 5 mm collimator using 220 kVp and 13 mA (Figure 1B). After 6 months, mice were euthanized and blood was collected through cardiac puncture. Additionally, irradiated liver tissue sections were flash-frozen in liquid nitrogen and stored at −80 °C or were fixed in formalin and paraffin embedded for further analysis. All radiation sham mice were anaesthetized and placed in the SARRP for an equivalent time as the irradiated treated mice. All animal protocols and procedures used in this study were approved (AUP# 3750) by the Institutional Animal Care and Use Committees of the University of Arkansas for Medical Sciences.

### 2.2. Immunohistochemistry and Histopathology Analysis

Sections were deparaffinized and rehydrated using decreasing concentrations of ethanol. One set of slides was stained with hematoxylin and eosin. These slides were then scored by a clinical pathologist to determine the level of liver injury in a double-blinded manner. Differences to the sham mice groups, when present, were noted in several categories including possible micro/macrovesicular steatosis, lymphoplasmacytic inflammation (i.e., portal, perivenular, and lobular regions), necrosis, fibrosis, angiectasis, and the presence of any regeneration nodules. Each liver section in each group was given a verbal score of “none”, “mild” and “moderate” that was translated into the table as “−, +, and ++”; the table also includes how many animals out of the group presented the liver injury marker in the group.

Another set of slides was stained for DNA damage using a fluorescent Terminal deoxynucleotidyl transferase (TdT) dUTP Nick-End Labeling (TUNEL) assay. Analysis of DNA damage was determined by a double-blinded imaging and scoring of 10 random 40X fields per section for positive (green) compared to total hepatocyte nuclei (blue).

For the immunohistochemical staining for 3-nitrotyrosine and Cytokeratin-19, the tissue slides were deparaffinized and endogenous peroxidase was quenched followed by incubation in Dako protein-block to block nonspecific binding. Anti-3-nitrotyrosine rabbit polyclonal antibody (Millipore; #06284, 1:1000) was applied for 1 h in Dako antibody diluent buffer. Rat Anti-Cytokeratin-19 antibody (DSHB Hybridoma Product TROMA-III; #ab2133570, 1:300) was incubated for two hours in Dako diluent. All sets were then incubated in Vector Biotinylated Goat Anti–Rabbit–1:400 prepared in TBS-T for 30 min. Then slides were incubated in Vector ABC Elite for 30 min. Slides were developed with Dako diaminobenzidine (DAB). Slides were counterstained with hematoxylin and mounted. The negative control slides followed the same protocol but did not use the primary antibody. 3-Nitrotyrosine immunohistochemical staining was quantified by counting positive cells near similar sized central veins (cytoplasmic or nuclear staining) per 400× field with the following scoring system: 0 (0 positive cells), 1 (1–20 positive cells), 2 (21–30 positive cells), 3 (31–40 positive cells) and 4 (>41 positive cells). A total of 15 × 400 × fields were scored, and means of these scores were calculated. Bile ducts were scored and counted in Cytokeratin-19 stained slides by examining 8-10 regions containing at least one portal area per liver section.

### 2.3. Real Time Quantitative Reverse Transcription PCR (qRT-PCR)

Total RNA was extracted from flash-frozen liver using the DNA/RNA/Protein extraction kit (IBI Scientific IB47702, Peosta, IA, USA) according to the manufacturer’s protocol. After extraction, the total RNA concentration was determined. cDNA was synthesized from 1000 ng RNA using the High-Capacity cDNA Reverse Transcription kit (Thermo Fisher Scientific, Waltham, MA, USA) as we previously described [47].

Fibrosis and inflammation marker levels of gene transcription were determined by single gene quantitative reverse transcription polymerase chain reaction (qRT-PCR) using Fast TaqMan Gene Expression Assays (Life Technologies, Carlsbad, CA, USA) and the Applied Biosystems 7500 Real Time PCR system. The genes of interest included: TGF-β, α-SMA, Procoll1, IL-6, and IL-1β; see Appendix A. The relative fold change of mRNA for each gene of interest was determined using the Comparative CT (2^−ΔΔCt^) method. These results were normalized to house-keeping gene GAPDH values [47].

### 2.4. Antioxidant Enzyme Activity Measurements

Whole flash-frozen liver tissue was homogenized in 50 mM potassium phosphate buffer containing 1.34 mM diethylenetriaminepentaacetic acid (pH: 7.4) on ice. Protein concentrations were determined using a Lowry’s Protein Assay as previously described in [48].

Catalase activity for liver homogenates was determined using a 1:10 dilution from the homogenate. Absorbance was read using the spectrophotometer at 240 nm for both the reference blank and the sample. 50 μL of the diluted sample was placed in a total of 4 mL of 50 mM (pH 7.0) phosphate buffer. Then 2 mL of this total solution was placed into a sample cuvette and a reference blank cuvette. The addition of 1 mL of 30 mM H_2_O_2_ into the sample cuvette began the reaction and absorbance change vs. time was measured for 2 min. The absorbance of the reference blank was subtracted from the sample cuvette to determine the level of activity per mg of protein as previously described [49].

Glutathione Peroxidase activity was calculated using an established protocol [50]. A 1:50 dilution of liver homogenate was added into a mixture of “working buffer” (containing 1.33 mM reduced glutathione (GSH), 1.33 U/mL glutathione reductase (GR) and 55.6 mM potassium phosphate buffer (pH: 7)) and 4 mM NADPH. This is incubated for 5 min at 30 °C, then the reaction is initiated when 15 mM cumene hydroperoxide is added. Absorption changes were measured at 340 nm for 3 min that was then compared to the blank absorbance for each sample cuvette. The activity is calculated in units per mg protein.

Glutathione Reductase activity was measured in a similar fashion to the Glutathione Peroxidase protocol. To initiate the reaction, a 1:10 dilution of liver homogenate was added to ddH_2_O, PB/EDTA (100 mM potassium phosphate buffer/3.4 mM EDTA), 30 mM oxidized glutathione (GSSG), 0.8 mM NADPH, and 1% bovine serum albumin (BSA). The decrease in absorbance at 340 nm was measured for 3 min and subtracted from the blank absorbance. The absorbance was used to calculate activity in units per mg protein as we previously described [51].

MnSOD activity was measured based on the protocol established by Spitz and Oberley [52]. It is determined by recording the rate of reduction of nitroblue tetrazolium (NBT) from the superoxide generated by xanthine oxidase. The superoxide dismutase (SOD) present in the samples will scavenge the superoxide generated. This is a competitive inhibition of the reduction of NBT. A unit of SOD is considered the amount of SOD required to inhibit 50% of the NBT reduction. This is determined using varying concentrations of sample to find the percent inhibition curve and determine the concentration at 50% inhibition. The MnSOD activity is determined by adding 0.33 M sodium cyanide buffer to inhibit the CuZnSOD enzyme activity.

### 2.5. Bilirubin Assay

Plasma bilirubin levels were measured following Bilirubin Assay Kit protocol (Sigma-Aldrich, St. Louis, MO, USA) using 50 μL of blood plasma in each well of 96 well plates to determine the total bilirubin, direct bilirubin and blank absorbance. Absorbance was measured at 530 nm. The indirect bilirubin levels were determined by subtracting the total bilirubin numbers from the direct bilirubin, which is also known as conjugated bilirubin.

### 2.6. Statistical Analysis

Statistical analysis was performed using GraphPad Prism 8.0 (GraphPad Software, San Diego, CA, USA). Data are expressed as mean ± 1SD, unless otherwise specified. One-way ANOVA analysis with Tukey’s post-analysis was used to study the differences among the study group’s means. Significance was determined at *p* < 0.05 and the 95% confidence interval. Statistical significance is expressed as * *p* < 0.05, ** *p* < 0.01, and *** *p* < 0.005.

## 3. Results

### 3.1. Exposure to 24 Gy IR Increased the Expression of Inflammatory Markers as Well as Lymphoplasmacytic Inflammation in the Livers of Sirt3^−/−^ Mice

The histology of the liver sections was scored in a double-blinded manner to determine the level of injury after irradiation. The scoring demonstrated considerably increased inflammatory cells in the Sirt3^−/−^ mice 6 months after 24 Gy IR, and while there were no severe cases yet, several animals displayed moderate inflammation in the portal and perivenular space of the murine livers (Figure 2).

We also measured the gene expression of inflammatory chemokines, IL-6, IL-1β, and TGF-β, in the livers of sham or 24 Gy treated mice from both Sirt3^+/+^ and Sirt3^−/−^ mice. Previous studies have demonstrated increased expression levels of these chemokines six months after liver-targeted irradiation. Additionally, IL-1β is an important chemokine in the formation of the liver’s inflammasome that leads to increased liver inflammation and tissue damage. These increases in inflammatory chemokines and profibrotic factors could contribute to the removal of necrotic tissue after irradiation and stimulate extracellular matrix deposits that assist in regeneration. However, the chronic release of IL-6, IL-1β, and TGF-β would result in increased liver damage and fibrosis. Expression of IL-6, IL-1β, and TGF-β were measured in triplicates for each animal in a group. The values were averaged and normalized to GAPDH as the housekeeping gene. Fold change was determined by comparing each group to its own sham. Only Sirt3^−/−^ mice, which received 24 Gy IR, showed a significant increased expression of these chemokines (Figure 3).

### 3.2. The Number of Bile Ducts Was Decreased in Irradiated Livers of Sirt3^−/−^ Mice

Liver tissue sections were stained with cytokeratin-19 to visualize and score the bile duct presence. Double-blind scoring showed a significant reduction of number of bile ducts per portal area in the Sirt3^−/−^ irradiated mice (Figure 4), which could contribute to the development of cholestasis and eventually chronic fibrosis. Interestingly there were no changes in the plasma bilirubin levels (conjugated or unconjugated; Appendix A) probably due to the small irradiation field that was used in the study.

### 3.3. Exposure to 24 Gy IR Increased the Expression of Profibrotic Markers in the Livers of Sirt3^−/−^ Mice but Did Not Cause Fibrosis at 6 Months after IR Exposure

Because TGF-β is a cytokine growth factor that signals the release of additional inflammatory chemokines and fibrotic growth factors, next we measured gene expression of procollagen-1 and α-SMA in the sham and 24 Gy irradiated livers of both Sirt3^+/+^ and Sirt3^−/−^ mice. Expression of these two profibrotic marker proteins was measured in triplicates for each animal in a group. The values were averaged and normalized to GAPDH as the housekeeping gene. Only Sirt3^−/−^ mice, which received 24 Gy IR, showed a significant increased expression of procollagen-1 and α-SMA (Figure 5). However, this increase did not translate into a bridging fibrosis (data not shown), which may develop in severe RILD patients.

### 3.4. Persistent Significant Increases in DNA Damage and Oxidative Stress in Liver Tissue of Sirt3^−/−^ Mice Following 24 Gy IR

DNA damage is an early event during exposure to IR. However, chronic inflammation and oxidative/nitrosative stress have been suggested to contribute long-term detrimental effects of radiation exposures by creating a feed-forward mechanism for persistent injuries to macromolecules, such as DNA. Since we have seen increased expression of inflammatory markers 6 months post IR in liver tissue of Sirt3^−/−^ mice, we determined the levels of protein oxidation using 3-nitrotyrosine immunohistochemistry. Blind scoring of liver sections were performed at 15 randomly selected central veins, and then given a score of 0 (0 positive cells), 1 (1–20 positive cells), 2 (21–30 positive cells), 3 (31–40 positive cells), or 4 (41 positive cells or greater). Our results showed that 24 Gy IR increased levels of protein oxidation in mouse livers, which were persistent months after the exposure (Figure 6).

Therefore, it was not completely surprising when Sirt3^−/−^ livers in irradiated group also exhibited a significant increase of TUNEL positive cells compared to sham irradiated mice, suggesting DNA fragmentation and possible apoptosis still exists in liver cells lacking SIRT3 as a stress response protein against IR-induced oxidative injury (Figure 7).

### 3.5. Mice Lacking SIRT3 Demonstrated Altered Activity of Antioxidant Enzymes, Which Are Responsible for Peroxide Removal in Liver Tissue Following 24 Gy IR

As a major deacetylation target of SIRT3, increased activity MnSOD has been shown to respond to exogenous cellular stressors including acute IR injury in liver tissue [29] and loss of SIRT3 has been associated with increased levels of superoxide [29,30,42]. Therefore, we first examined whether MnSOD activity has not been altered in irradiated Sirt3^−/−^ mice but it was increased in Sirt3^+/+^ mice, via deacetlyation of critical lysine residues at its active site. No significant changes were noted in liver MnSOD activity between the sham and irradiated groups in either genotype (Figure 8).

Interestingly, the activity of antioxidant enzymes CAT and GPx were significantly higher in irradiated Sirt3^−/−^ mice, while GR activity was significantly lower in this group compared to its sham irradiated counterparts (Figure 9).

This result suggested that peroxide mediated oxidative injury is more relevant in radiation-induced long-term liver toxicity.

## 4. Discussion

Almost half of all men and women have a lifetime probability of developing a new invasive cancer; therefore, radiation therapy and chemotherapy are being used more frequently [1]. The liver, located in the abdominal region, is a large organ that plays a critical role in metabolic homeostasis and detoxification. Because of this metabolic role, the liver contains about 25% of mitochondria in its cytosolic space [20]. This high mitochondrial content and oxygen requirement makes the liver especially susceptible to radiation-induced ROS that can lead to hepatic radiation-induced damage, even with doses as low as 30 Gy [10,19]. The typical clinical presentation of RILD develops 3 months after irradiation and is associated with increased abdominal girth, ascites, hepatomegaly, and veno-occlusive disease that can eventually progress to liver failure [4].

Although clinical observations are established and patient-oriented morphological characteristics of IR-induced pathologies are well described, we have limited understanding of molecular mechanisms that would explain dose-limiting toxicities in patients treated with high-dose IR [3,4,7,13,17,18,53,54,55,56]. In addition, no pharmacological interventions have been shown to be consistently efficacious [9,54]. Therefore, understanding the confounding factors and mechanisms involved in IR-induced long-term liver pathologies will particularly benefit the high-risk populations with underlying liver dysfunction. The lack of prevention and therapy post RILD diagnosis exacerbate the need for pre-clinical models that will allow the accurate study of radiation-induced liver damage development [2].

IR induces not only immediate production of free-radical mediated effects but also persistent increases in metabolic production of reactive species, which contribute to the long-term tissue effects of radiation [57]. The damage response cascades (i.e., inflammation, necrosis and fibrosis) are further stimulated from the consistent formation of reactive species in days or even months after initial irradiation exposure [19,24,58]. Similar damage response cascades were seen in our Sirt3^−/−^ model (Figure 2, Figure 3, Figure 4, Figure 5, Figure 6 and Figure 7) 6-months after 24 Gy radiation was delivered to a small (5 × 5 mm) field of the liver indicating persistent oxidative injury via the indirect generation of ROS and RNS. Superoxide anion, hydrogen peroxide, hydroxyl radical, and peroxynitrite are a few examples of the radiation-induced reactive species which can cause mitochondrial alterations that lead to oxidative and nitrosative stress signaling and genomic instability; all adding to the long-term injury and dysregulation of normal liver parenchymal function [59,60]. ROS and RNS can be removed through several different antioxidant mechanisms, including SODs, CAT, and the glutathione, thioredoxin, and peroxiredoxin systems, most of which are regulated via transcriptional and/or post-translational modifications [29,30,31,32,33,34]. SIRT3, a member of NAD+ dependent enzymes family, has been shown to increase the activity of antioxidant enzymes, including MnSOD [30,31,32,33,34,35,36,37,38,39,40,41,42,43,44,45,46,47,48,61]. In agreement with these studies, Coleman et al. demonstrated SIRT3’s stress response function in an acute radiation-induced liver injury model, in which the authors provided evidence for O_2_^•−^ mediated liver injury in the absence of SIRT3, 48 h after 4 Gy total body irradiation exposure [29]. Interestingly, in our current study of IR-induced long-term liver injury, the model suggested that O_2_^•−^ may no longer be the major participant in the chronic liver damage, as we did not see any changes in enzymatic activity of MnSOD in the presence or absence of SIRT3 6-months following targeted irradiation (Figure 8).

After examination by a clinical pathologist, liver histology of the irradiated fields exhibited a moderate increase in lymphoplasmacytic inflammation in the perivenular space, implying an increased inflammation that can further signal of tissue necrosis or fibrosis. In an acute response to irradiation or other types of liver injury, increased inflammation is necessary for tissue regeneration. Inflammation facilitates a fibrotic network for regrowth and remodeling of liver tissue as well as vascular network using chemokines like TGF-β, IL-1β, and IL-6. However, continual elevations of inflammatory cells releasing, secreting and recruiting more cells using cytokines like TGF-β, can be associated with the development of chronic liver disease [62]. While IL-1β and IL-6 are initially released to protect the surrounding tissue from further damage in an acute response, its persistent and continued exposure causes recruitment of additional Kupffer cells within the liver that triggers a feed-forward cycle of continuous inflammation, damaging the hepatocytes [63]. In our targeted irradiation model, we found that there was a significant increase in TGF-β, IL-1β, and IL-6 mRNA expression, suggesting a persistent inflammatory response and continuous liver injury progression 6 months after treatment in the absence of SIRT3. This fold increase in the Sirt3^−/−^ mice was significantly higher than in the Sirt3^+/+^ irradiated group, suggesting that the release of IL-6 and IL-1β are no longer a beneficial injury response due to its new sustained nature. The elevation in inflammation cytokines and histochemistry corroborate the idea that the inflammation is chronic in our model and will eventually lead to further liver toxicity, demonstrating that functional SIRT3 is important for liver healing and the balance between inflammation and regeneration even 6 months after irradiation. Adding to this finding, the increased TUNEL staining confirms DNA degradation in the irradiated Sirt3^−/−^ mice, further signifying the development of long-term hepatocyte injury from persistent inflammatory cascades. These results confirm the protective role of SIRT3, which was previously demonstrated in a sepsis model by describing its molecular links to DNA damage, apoptosis and inflammatory responses via NLRP3 inflammasome upregulation as well as apoptosis-associated speck-like protein in the absence of functional SIRT3 [64].

Sinusoidal obstruction and hepatic fibrosis are classical features of radiation-induced liver injury in patients. Histology of the samples in the current study did not show any veno-occlusive events or an increase in fibrotic collagen staining. However, hepatobiliary tract toxicities including vanishment of bile ducts and elevated alkaline phosphatase are also common developments after radiation exposures, thus we used cytokeratin 19 staining for the visualization of bile ducts. The results demonstrate an interesting finding of possible bile ductopenia in the Sirt3^−/−^ mice 6 months after IR exposure, which was not observed in wild-type irradiated mice. Although to date no direct mechanistic link has been established between SIRT3 function and bile duct injury, literature evidence suggests a strong correlation between increased oxidative stress and increased levels of apoptosis in biliary epithelial cells, which was mediated by glutathione in primary biliary cirrhosis [65]. Furthermore, increased inflammatory cytokine-induced oxidative stress has been shown to decrease the expression of biliary markers through miRNA506 regulation of DNA damage and apoptosis [66]. Corroborating with the findings of these studies, our results also revealed increased levels of 3-nitrotyrosine on irradiated Sirt3^−/−-^ livers (Figure 6), which evidenced the existence of sustained oxidative injury in irradiated Sirt3^−/−^ livers. While the irradiated liver sections obtained from both genotypes did not demonstrate an increase in fibrotic collagen staining (data not shown), based on our findings with decreased number of functioning bile ducts in Sirt3^−/−^ mice and elevated mRNA of fibrotic factors (α-SMA and procollagen 1), we believe these results indicate that these mice will eventually develop persistent fibrosis in the irradiated field [67].

The morphological evidence we presented in our current study strongly suggests a noteworthy role SIRT3 plays in the long-term radiation-induced liver toxicity. As an NAD+ dependent deacetylase, which resides primarily in mitochondria, SIRT3 functions to maintain redox homeostasis under stress. A total body irradiation study by Coleman et al., demonstrated that 48 h after exposure, there is an early increase in O_2_^•−^ generation in Sirt3^−/−^ mice livers, which mediated the increased acute liver injury in these animals [29]. We initially expected to see a similar response in MnSOD activity in the long-term progression of radiation-induced injury. However, there was no significant change in MnSOD activity between sham and irradiated groups at six months after irradiation (Figure 8). Because MnSOD, and thus O_2_^•−^ were not likely to be involved in the IR-induced chronic liver injury at 6 months time point, we shifted our focus to other antioxidant networks that may now be attempting to overcome the oxidative stress from radiation.

As a number of SIRT3 deacetylation targets occur in mitochondria, which could indirectly contribute to the redox homeostasis by providing the necessary coenzyme NADPH (e.g., isocitrate dehydrogenase 2) [31,32], we examined alternate antioxidant enzymes. Our data indicate a possible switch from O_2_^•−^ to hydroperoxide metabolism that contributes to the injury endpoints found in this study. In the absence of SIRT3, 6 months after radiation there was a significant decrease in GR. This is an important enzyme that requires NADPH for its enzymatic activity for glutathione recycling. Without SIRT3 there may have been a decrease in availability of the necessary coenzyme NADPH for the continued recycling of glutathione in response to the persistent elevated oxidative stress generated from radiation exposure.

Considering that peroxiredoxin 3, the principal peroxidase responsible for mitochondrial hydrogen peroxide, is also a SIRT3 deacetylation target [68], accumulation of hydroperoxides and hydrogen peroxide in irradiated livers resulted in the significant increases in GPx and CAT activities we measured in Sirt3^−/−^ mice. However, with long-term elevations clearly still seen after 6 months, this antioxidant system also appears to be overwhelmed in the absence of SIRT3, leading to the liver toxicity seen in our study.

## 5. Conclusions

In conclusion, our results exhibited a vital function of SIRT3, a major mitochondrial deacetylase, in IR-induced long-term liver injury. Loss or decrease in SIRT3 levels could be an underlying factor and contributor to a damage-permissive phenotype in murine liver long after exposure to IR. Although our model did not fully represent the clinical indications of human RILD, we believe this was due to the small volume of irradiation chosen in the design. There were important signs that this murine model may relate to the clinical presentation of human RILD including ductopenia, lymphoplasmacytic inflammation, continued hepatocyte toxicity from DNA damage and elevations in the expression of fibrotic factors.

The data presented in the current study also strongly suggested that O_2_^•−^ driven acute liver injury following IR exposure appears to shift towards a peroxide-mediated long-term injury and clearly is no longer limited to the mitochondria. Thus, specific SIRT3 target proteins and the reactive species contributing to the progression of RILD, merit investigation to establish causal mechanisms for IR-induced chronic liver toxicity and develop strategies to prevent RILD in cancer survivors.

## Figures and Tables

**Figure 1 antioxidants-09-00409-f001:**
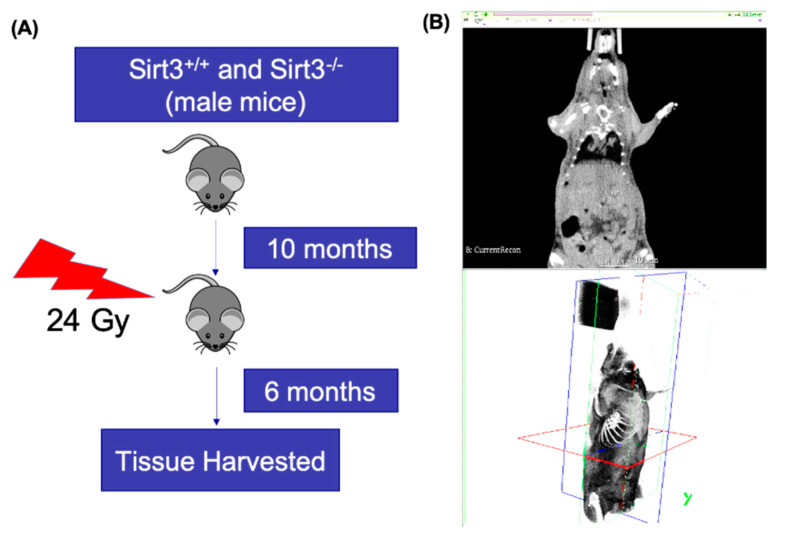
(**A**) Experimental timeline for the irradiation and tissue harvest from Sirt3^+/+^ and Sirt3^−/−^ male mice. (**B**) Image guided irradiation of the liver using Small Animal Radiation Research Platform (SARRP).

**Figure 2 antioxidants-09-00409-f002:**
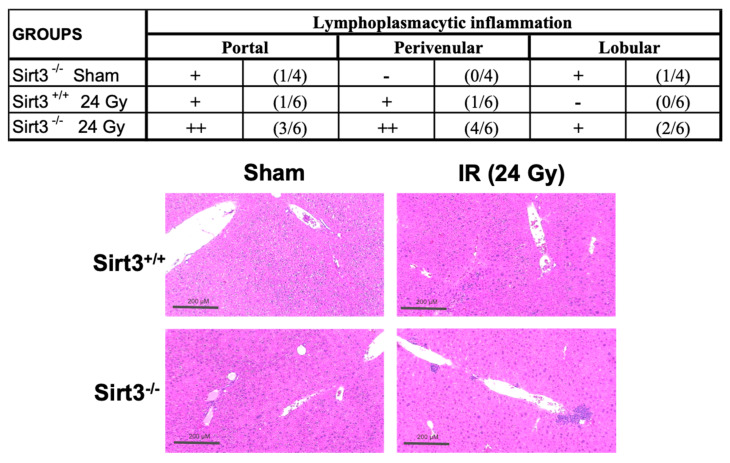
Increased numbers of inflammatory cells were seen in irradiated Sirt3^−/−^ mice. H&E scoring: − (none), + (mild/focal), and ++ (moderate). All sections were normalized to the Sham Sirt3^+/+^ group and presented as number of animals with marker per animals in each group (*n* = 4–6). Sham groups underwent the same procedures without receiving 24 Gy IR (irradiation).

**Figure 3 antioxidants-09-00409-f003:**
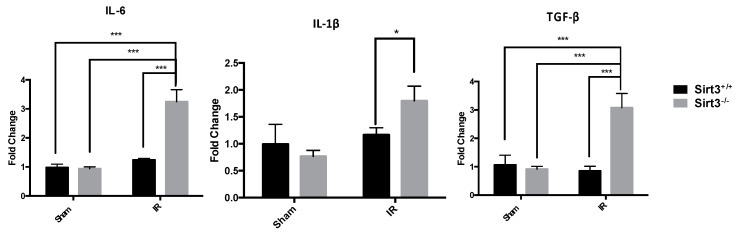
Expression of inflammatory markers determined by quantitative real-time PCR. IL6 and TGFβ were significantly increased in Sirt3^−/−^ mice after irradiation compared to sham irradiated Sirt3^+/+^ or Sirt3^−/−^ as well as Sirt3^+/+^ irradiated groups. IL1β was significantly increased in irradiated Sirt3^−/−^ mice compared to irradiated Sirt3^+/+^ livers (*n* = 4–6; * *p* < 0.05, *** *p* < 0.001).

**Figure 4 antioxidants-09-00409-f004:**
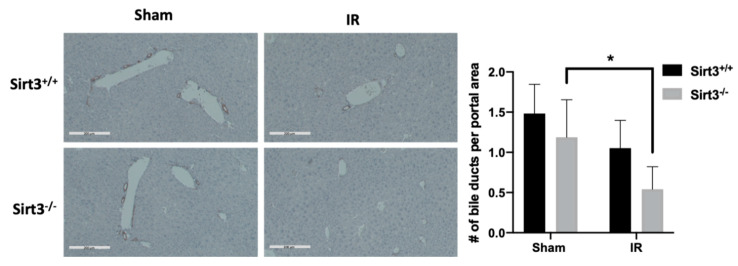
Sirt3^−/−^ irradiated mice showed decreased number of bile ducts. Liver sections were stained and counted with anti-cytokeratin 19 to determine the presence of bile ducts. Scale bars: 200 µm. (8–10 fields were scored per mouse, *n* = 6; * *p* < 0.01).

**Figure 5 antioxidants-09-00409-f005:**
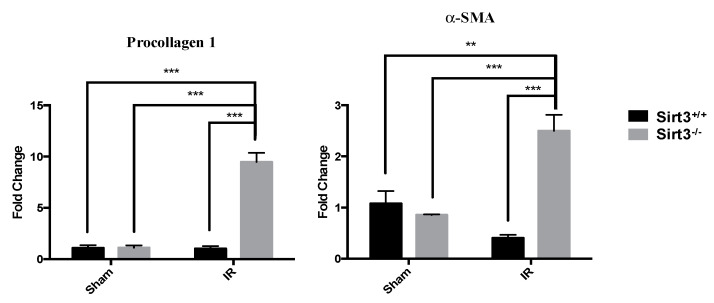
Expression of profibrotic markers were increased in all Sirt3^−/−^ mice after irradiation compared to sham irradiated Sirt3^+/+^ or Sirt3^−/−^ as well as Sirt3^+/+^ irradiated groups (*n* = 4–6; ** *p* < 0.01, *** *p* < 0.001).

**Figure 6 antioxidants-09-00409-f006:**
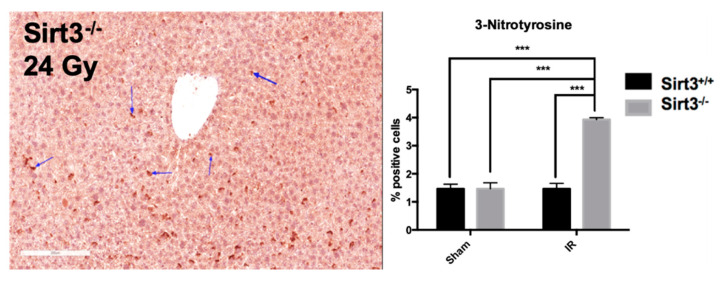
Protein oxidation, determined by 3-nitrotyrosine staining (indicated by the arrows), was increased in irradiated Sirt3^−/−^ mice compared to sham irradiated Sirt3^+/+^ or Sirt3^−/−^ as well as Sirt3^+/+^ irradiated groups. Scale bar: 200 µm (15 fields scored per mouse, *n* = 4–6; *** *p* < 0.001).

**Figure 7 antioxidants-09-00409-f007:**
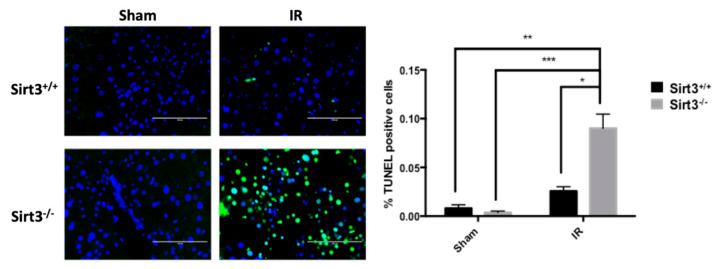
The number of TUNEL positive cells were significantly increased in Sirt3^−/−^ mice after irradiation compared to sham irradiated Sirt3^+/+^ or Sirt3^−/−^ as well as Sirt3^+/+^ irradiated groups. Scale bar: 100 µm (*n* = 4–6; * *p* < 0.05 ** *p* < 0.01, *** *p* < 0.001).

**Figure 8 antioxidants-09-00409-f008:**
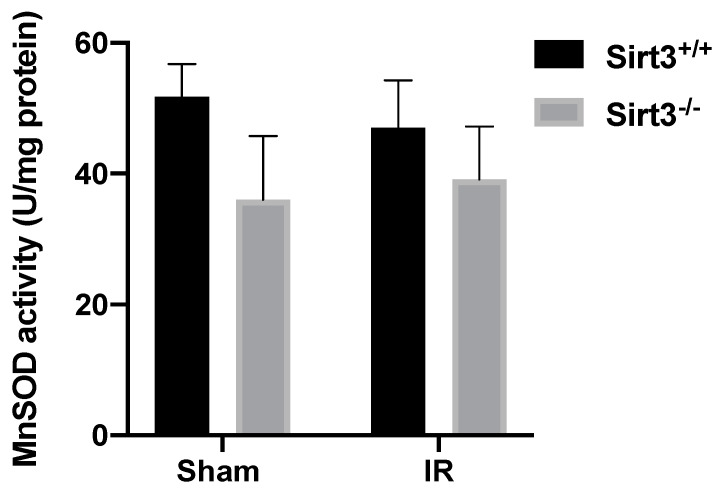
Exposure to 24 Gy liver only irradiation did not significantly alter enzymatic activity of MnSOD in Sirt3^+/+^ or Sirt3^−/−^ mice (*n* = 4–6).

**Figure 9 antioxidants-09-00409-f009:**
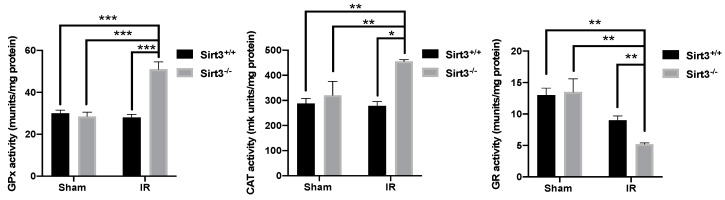
Glutathione Peroxidase (GPx) and catalase (CAT) activity were significantly increased while Glutathione Reductase (GR) enzymatic activity was decreased in irradiated Sirt3^−/−^ mice compared to sham irradiated Sirt3^+/+^ or Sirt3^−/−^ as well as Sirt3^+/+^ irradiated groups (*n* = 4–6; * *p* < 0.05, ** *p* < 0.01, *** *p* < 0.001).

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
