# Peer review of "The Role of Sirtuin 3 in Radiation-Induced Long-Term Persistent Liver Injury"

_antioxidants, 2020, doi:10.3390/antiox9050409_

Round 1
Reviewer 1 Report
In this study the authors evaluated the effect radiation (24 Gy) in liver of ten-month-old Sirt3-/- and Sirt3+/+ male mice. They performed immunoistochemical and qPCR analyses and antioxidant enzyme activities at 6-month time point following radiation exposure. They found significant increase in expression of inflammatory and profibrotic markers in the irradiated areas of Sirt3-/- livers compared to their sham treated controls. They did not find significant changes in MnSOD activity in sham versus irradiated groups in neither Sirt3-/- nor Sirt3+/+ mice.
The study is potentially interesting however the manuscript is written in a confused way and the conclusion are not evidenced.
The authors should rewrite the manuscript following a more logic way for planned experiments. The authors should definitively rewrite the Discussion part that, as it is, looks like a review instead. The authors should clearly state which is the main finding of the study and discuss it according to the literature.
The authors should try to explain the molecular link between SIRT3-/- and increased inflammatory markers IL-6, IL-1β, and TGF-β,
between SIRT3-/- and the number of bile ducts
between SIRT3-/- and the increase of profibrotic markers procollagen-1 and α-237 SMA
between SIRT3-/- and DNA damage and oxidative stress.
As it is, the manuscript is just a list of experiments without any link among them.
Author Response
REVIEWER 1
The study is potentially interesting however the manuscript is written in a confused way and the conclusion are not evidenced. The authors should rewrite the manuscript following a more logic way for planned experiments.
1) The authors should definitively rewrite the Discussion part that, as it is, looks like a review instead. The authors should clearly state which is the main finding of the study and discuss it according to the literature.
We agree with the reviewer and extensively revised the text to present and discuss our findings in a more logical fashion.
2) The authors should try to explain the molecular link between SIRT3-/- and increased inflammatory markers IL-6, IL-1β, and TGF-β, between SIRT3-/- and the number of bile ducts, between SIRT3-/- and the increase of profibrotic markers procollagen-1 and α-SMA, between SIRT3-/- and DNA damage and oxidative stress. As it is, the manuscript is just a list of experiments without any link among them.
Our revised discussion now links the loss of SIRT3 and our data demonstrating changes in bile ducts, inflammatory as well as profibrotic markers, DNA and oxidative damage, according to the recent literature (new references [64, 65, 66, 68] were included). We discussed how loss of functional SIRT3 could be an underlying risk factor due to its link to these parameters (inflammation, DNA damage, oxidative stress), which have been widely accepted as mechanisms for radiation induced chronic liver injury.

Reviewer 2 Report
It is an interesting article with significant changes in the measured parameters supporting the main hypothesis.
However, some changes are needed:
Address of the authors is incomplete
Abstract is incomplete, states the objectives and part of the methodology but no results or conclusions are shown. Therefore it should be completely rewritten.
Sirt role in liver radiation damage has been clearly stated: Lack of novelty, a very similar approach is already published on “Mitchell C. Coleman, Alicia K. Olivier, James A. Jacobus, Kranti A. Mapuskar, Gaowei Mao, Sean M. Martin, Dennis P. Riley, David Gius, and Douglas R. Spitz. Antioxidants & Redox Signaling.Mar 2014.1423-1435. http://doi.org/10.1089/ars.2012.5091.
However, the important point in this article is the quantitation of the long term effects of the irradiation in the animals. Abstract and discussion should be modified to stress this point.
Also if any data of short term effects of the irradiation on the animals could be provided, it would greatly improved the manuscript. Also, from this point of view a more detailed discussion to justify the MnSOD activity should be incorporated. Furthermore this lack of changes should be explained since a significant modification in other antioxidant enzymes is detected.
In figure 2, images of all experimental groups should be included.
Author Response
REVIEWER 2
It is an interesting article with significant changes in the measured parameters supporting the main hypothesis. However, some changes are needed:
1) Address of the authors is incomplete.
The address was completed.
2) Abstract is incomplete, states the objectives and part of the methodology but no results or conclusions are shown. Therefore, it should be completely rewritten.
Abstract was rewritten to summarize the major findings and conclusions of the study
3) Sirt role in liver radiation damage has been clearly stated: Lack of novelty, a very similar approach is already published on “Mitchell C. Coleman, Alicia K. Olivier, James A. Jacobus, Kranti A. Mapuskar, Gaowei Mao, Sean M. Martin, Dennis P. Riley, David Gius, and Douglas R. Spitz. Antioxidants & Redox Signaling.Mar 2014.1423-1435. http://doi.org/10.1089/ars.2012.5091. However, the important point in this article is the quantitation of the long term effects of the irradiation in the animals. Abstract and discussion should be modified to stress this point.
We agree with the reviewer for his/her comment regarding to emphasize the differences between Coleman et al., and our study. Abstract and discussion sections were extensively revised/re-written to stress the differences between the study by Coleman et al (Short term effects of total body irradiation of 4Gy in liver) and the current work (Long term effects of targeted 24 Gy irradiation in liver tissue).
4) If any data of short term effects of the irradiation on the animals could be provided, it would greatly improved the manuscript. Also, from this point of view a more detailed discussion to justify the MnSOD activity should be incorporated. Furthermore, this lack of changes should be explained since a significant modification in other antioxidant enzymes is detected.
We agree with the reviewer that any data from acute effects of 24 Gy exposure targeted to liver in these mice would be quite informative. However, considering the model used in our study (age 10 month old cohorts at the time of radiation exposure), we cannot provide such data in the time allotted for the revised manuscript submission. Revised text discusses why we initially focused on the MnSOD activity and when we did not see any changes, we shifted our focus on other antioxidant enzymatic networks which might also be negatively affected in the absence of SIRT3, especially when they are challenged with external stressors like radiation. Our future studies will continue to investigate the progression of liver pathologies we observed in the current work, using larger irradiated liver volume as well as collecting tissue at multiple time points following radiation exposure (24 h – 9 months) in order to mechanistically link the SIRT3 deacetylation targets, which could contribute to both superoxide and hydroperoxide mediated tissue injuries at different stages of RILD.
5) In figure 2, images of all experimental groups should be included.
Images of all experimental groups are included in Figure 2 in the revised manuscript.
RESPONSE TO THE REVIEWERS’ COMMENTS
REVIEWER 1
The study is potentially interesting however the manuscript is written in a confused way and the conclusion are not evidenced. The authors should rewrite the manuscript following a more logic way for planned experiments.
1) The authors should definitively rewrite the Discussion part that, as it is, looks like a review instead. The authors should clearly state which is the main finding of the study and discuss it according to the literature.
We agree with the reviewer and extensively revised the text to present and discuss our findings in a more logical fashion.
2) The authors should try to explain the molecular link between SIRT3-/- and increased inflammatory markers IL-6, IL-1β, and TGF-β, between SIRT3-/- and the number of bile ducts, between SIRT3-/- and the increase of profibrotic markers procollagen-1 and α-SMA, between SIRT3-/- and DNA damage and oxidative stress. As it is, the manuscript is just a list of experiments without any link among them.
Our revised discussion now links the loss of SIRT3 and our data demonstrating changes in bile ducts, inflammatory as well as profibrotic markers, DNA and oxidative damage, according to the recent literature (new references [64, 65, 66, 68] were included). We discussed how loss of functional SIRT3 could be an underlying risk factor due to its link to these parameters (inflammation, DNA damage, oxidative stress), which have been widely accepted as mechanisms for radiation induced chronic liver injury.
REVIEWER 2
It is an interesting article with significant changes in the measured parameters supporting the main hypothesis. However, some changes are needed:
1) Address of the authors is incomplete.
The address was completed.
2) Abstract is incomplete, states the objectives and part of the methodology but no results or conclusions are shown. Therefore, it should be completely rewritten.
Abstract was rewritten to summarize the major findings and conclusions of the study
3) Sirt role in liver radiation damage has been clearly stated: Lack of novelty, a very similar approach is already published on “Mitchell C. Coleman, Alicia K. Olivier, James A. Jacobus, Kranti A. Mapuskar, Gaowei Mao, Sean M. Martin, Dennis P. Riley, David Gius, and Douglas R. Spitz. Antioxidants & Redox Signaling.Mar 2014.1423-1435. http://doi.org/10.1089/ars.2012.5091. However, the important point in this article is the quantitation of the long term effects of the irradiation in the animals. Abstract and discussion should be modified to stress this point.
We agree with the reviewer for his/her comment regarding to emphasize the differences between Coleman et al., and our study. Abstract and discussion sections were extensively revised/re-written to stress the differences between the study by Coleman et al (Short term effects of total body irradiation of 4Gy in liver) and the current work (Long term effects of targeted 24 Gy irradiation in liver tissue).
4) If any data of short term effects of the irradiation on the animals could be provided, it would greatly improved the manuscript. Also, from this point of view a more detailed discussion to justify the MnSOD activity should be incorporated. Furthermore, this lack of changes should be explained since a significant modification in other antioxidant enzymes is detected.
We agree with the reviewer that any data from acute effects of 24 Gy exposure targeted to liver in these mice would be quite informative. However, considering the model used in our study (age 10 month old cohorts at the time of radiation exposure), we cannot provide such data in the time allotted for the revised manuscript submission. Revised text discusses why we initially focused on the MnSOD activity and when we did not see any changes, we shifted our focus on other antioxidant enzymatic networks which might also be negatively affected in the absence of SIRT3, especially when they are challenged with external stressors like radiation. Our future studies will continue to investigate the progression of liver pathologies we observed in the current work, using larger irradiated liver volume as well as collecting tissue at multiple time points following radiation exposure (24 h – 9 months) in order to mechanistically link the SIRT3 deacetylation targets, which could contribute to both superoxide and hydroperoxide mediated tissue injuries at different stages of RILD.
5) In figure 2, images of all experimental groups should be included.
Images of all experimental groups are included in Figure 2 in the revised manuscript.
Round 2
Reviewer 1 Report
The authors addressed the reviewer's comments and chenged the text accordingly.
Reviewer 2 Report
Albeit some expreriments that were requested have not been included, the manuscript has been greatly improved. Abstract and discussion are now focuss into the new findings presented in the manuscript. These results are properly reported and discussed.